# Feasibility of Unobtrusively Estimating Blood Pressure Using Load Cells under the Legs of a Bed

**DOI:** 10.3390/s24010096

**Published:** 2023-12-24

**Authors:** Gary Garcia-Molina

**Affiliations:** 1Sleep Number Labs, San Jose, CA 95113, USA; gary.garciamolina@sleepnumber.com or gmgarcia@wisc.edu; 2Center for Sleep and Consciousness, Department of Psychiatry, University of Wisconsin-Madison, Madison, WI 53719, USA

**Keywords:** load cells, bed, blood pressure, waveform reconstruction, unobtrusive sensing, deep learning

## Abstract

The ability to monitor blood pressure unobtrusively and continuously, even during sleep, may promote the prevention of cardiovascular diseases, enable the early detection of cardiovascular risk, and facilitate the timely administration of treatment. Publicly available data from forty participants containing synchronously recorded signals from four force sensors (load cells located under each leg of a bed) and continuous blood pressure waveforms were leveraged in this research. The focus of this study was on using a deep neural network with load-cell data as input composed of three recurrent layers to reconstruct blood pressure (BP) waveforms. Systolic (SBP) and diastolic (DBP) blood pressure values were estimated from the reconstructed BP waveform. The dataset was partitioned into training, validation, and testing sets, such that the data from a given participant were only used in a single set. The BP waveform reconstruction performance resulted in an R^2^ of 0.61 and a mean absolute error < 0.1 mmHg. The estimation of the mean SBP and DBP values was characterized by Bland–Altman-derived limits of agreement in intervals of [−11.99 to 15.52 mmHg] and [−7.95 to +3.46 mmHg], respectively. These results may enable the detection of abnormally large or small variations in blood pressure, which indicate cardiovascular health degradation. The apparent contrast between the small reconstruction error and the limit-of-agreement width owes to the fact that reconstruction errors manifest more prominently at the maxima and minima, which are relevant for SBP and DBP estimation. While the focus here was on SBD and DBP estimation, reconstructing the entire BP waveform enables the calculation of additional hemodynamic parameters.

## 1. Introduction

Cardiorespiratory signals such as heart rate, heart rate variability, respiratory rate, and blood pressure show distinct patterns between sleep and wakefulness states. Respiratory and heart rates progressively fall in the 5 to 10% range from sleep onset throughout a sleep session, and blood pressure (BP) falls by ~20%, reaching a minimum between 1.5 to 2.5 h after sleep onset [1]. BP decrease during sleep is referred to as dipping [2]. A nighttime BP decrease exceeding 20% (extreme dipping) may be associated with an increased risk for ischemic stroke and silent cerebral diseases. A BP decrease lower than 10% (non-dipping) or an increase in nighttime BP relative to daytime BP (reverse dipper or riser) is associated with an increased risk of death, myocardial infarction, and stroke [2].

To identify BP changes during sleep compared to waking hours, continuous and preferably cuffless BP monitoring is required. The use of a cuff-based BP measurement instrument may disturb sleep and defeat the purpose of measuring sleep-related BP changes.

Pulse transit time (PTT) [3], which can be estimated from measured electrocardiograms and photoplethysmograms (PPGs) as the interval duration between the ECG peak and the PPG peak or the PPG foot, is a valid and well-accepted estimate of BP [4]. Gesche et al. [5] used this approach and found Bland–Altman limits of agreement in the ±19.8 mmHg range to estimate instantaneous blood pressure.

However, reliably monitoring both ECG and PPG during sleep requires wearing wearables, which may interfere with sleep and may not be easily adhered to by the average user. Using PPG only, Slapni et al. [6] achieved mean absolute errors of 9.43 mmHg for systolic and 6.88 mmHg for diastolic BP using a spectro-temporal deep neural network.

Ballistocardiography (BCG) is a passive sensing technology which enables the measurement of motion, position changes, breathing, and small movements within the body, such as those generated by the ejection of blood with each heartbeat [7]. BCG is well suited to monitor sleep states as well as cardiac and respiratory activity during sleep as it does not require direct contact with the user’s body and can provide accurate results [8]. Multiple technology solutions exist to extract BCG signals [9], including electromechanical films, strain gauges, hydraulic sensors, fiber-optic sensors, and force (load cell) sensors.

Algorithms to estimate blood pressure using BCG data have been proposed in the literature. Seok et al. [10] used a sofa-style chair equipped with electro-film type sensors to derive two BCG signals whose phase difference was processed by a convolutional neural network, which resulted in approximate limits of agreement in the [−14 to +11 mmHg] and [−15 to +7 mmHg] range for the estimation of systolic and diastolic blood pressure, respectively. Chen et al. [11] used a BCG signal derived from an optical signal and combined it with a jointly recorded PPG to estimate their time delay and derive systolic and diastolic blood pressure with a mean standard error of 9 mmHg (standard deviation 1.8 mmHg). Kim et al. 2018 [12] used BCG signals from a force-plate and estimated diastolic and systolic blood pressure with respective limits of agreement in the [−14.6 to +14.6 mmHg] and [−17.5 to +17.5 mmHg] range. Tian et al. 2023 [13] used piezo-electric-derived BCG signals to obtain diastolic and systolic blood pressure with root-mean-square errors of 3.22 and 3.50 mmHg.

A recent publication by Carlson et al. [14] presented a concept aimed at enabling the tracking of cardiovascular parameters such as heart rate, stroke volume, and blood pressure using a bed equipped with multiple sensors, among which four load-cell sensors were placed under each leg of a bed. A load cell is a form of steel housing that contains a Wien bridge network that produces a differential voltage when a force is applied. A load-cell sensor can acquire movement and BCG data as well as static load data that indicate the center of the position of a person lying on the bed. Carlson et al. created a publicly available dataset [15] to promote algorithmic and scientific advances.

The goal of the research described in this article was to use the data in [15] and present feasibility results for the reconstruction of the continuous blood pressure waveform using the data from the four load cells. From the reconstructed blood pressure waveform, systolic and diastolic blood pressure were estimated.

## 2. Materials and Methods

### 2.1. Dataset

The data from 40 participants are available in IEEE DataPort at doi:10.21227/77hc-py84 [15] and include time-synchronized BCG signals from four load cells (TE Connectivity Measurement Specialties FX1901-0001-0200-L) and continuous blood pressure measured with a Finapres Medical System Finometer PRO. The Finometer PRO uses tonometry to continuously monitor arterial blood pressure via a small cuff placed around the finger.

The recordings lasted for 8.02 min on average across participants (see Table 1). Additional details concerning these data can be found in [14]. The location of the four load cells (LC0, LC1, LC2, and LC3) is shown in the top left portion in Figure 1. LC0 and LC3 are located on the bed’s head, while LC1 and LC2 are located on the bed’s foot.

Because of the measurement errors described in [15], the data from 2 participants out of the 40 failed to collect useful load-cell signals from LC0. For the purposes of this research, that data were not included.

Demographic and relevant health information for participants in the study are shown in Table 1.

### 2.2. Signal Acquisition and Processing

The diagram in Figure 1 shows an overview of signal processing and machine learning methods. Analog signals from each load cell were acquired at a sampling frequency of 1000 Hz, using a NI9220 analog input module controlled by a National Instruments—LabVIEW^TM^ v14.01 virtual instrument running on a computer. The signals were amplified and bandpass-filtered between 0.05 and 35 Hz. The blood pressure signal from the Finometer PRO was interfaced with a second NI9220 module. The LabVIEW^TM^ virtual instrument was configured to synchronously sample all the signals. Additional details regarding data acquisition can be found in [14,15].

The load-cell and blood pressure signals were subsampled from 1000 Hz to 100 Hz. The subsampling frequency was chosen given the power spectrum density content of the signals, depicted in Figure 2, which shows a decrease by several orders of magnitude in the spectral content at 50 Hz.

### 2.3. Blood Pressure Signal Reconstruction Algorithm

The reconstruction of the blood pressure signal was modeled as a regression problem where the objective was to estimate the blood pressure sample BP(*N*) at time *N* using a 4 × *N* matrix M(*N*) containing columns [LC0(*i*), LC1(*i*), LC2(*i*), LC3(*i*)], where LCj(*i*) is the *i*-th sample of the *j*-th load-cell signal resampled at 100 Hz for *i* = 0, …, *N*. Parameter *N* was set to 50, i.e., 500 milliseconds. Repeating this process through successive 1-sample shifts of matrix M enables the reconstruction of the entire BP waveform at 100 Hz except for the initial *N* samples.

The function to map the 4 × 50 matrix M(*N*) to the scalar value BP(*N*) was implemented as a neural network composed of a sequence of three 30-unit short-term memory (LSTM) [16] recurrent neural networks followed by a fully connected (dense) neural network. The specific input/output dimensions for each layer can be seen in Figure 3. This type of neural network architecture was inspired by earlier work by the author [17,18] on deep neural networks for physiological signal analysis.

### 2.4. Neural Network Training, Validation, and Testing Phases

A random permutation of the 38 participant identifiers P1,…, P38 was generated. The data from recordings of participants P1 to P22 corresponded to approximately 60% (22/38) of the data, P23 to P28 corresponded to 15% (6/38), and P29 to P38 corresponded to 25% (10/38) of the data used for training, validation, and testing, respectively. This partition of the data ensured that the samples from a given participant could only be used for either the training, validation, or testing phase (see Figure 3 for illustration).

The number of epochs for training and validation of the neural network was set to 300, and the batch size to 256. An epoch corresponds to the pass of the entire training data through the network, and the batch size is the number of pairs {M(*N*), BP(*N*)} that are used to estimate the error gradient. The mean squared error was set as a loss function, and the optimization method was the Adams algorithm [19]. There were 18,871 trainable parameters which were initialized according to the Glorot normal initialization strategy [20]. On completion of each epoch, the mean squared error for the training and validation sets was calculated (see Figure 3a,b).

The neural network was implemented in Python 3.7.16 using TensorFlow 2.3.0 and Keras 2.4.0 [21,22]. The training process was performed with a checkpoint callback such that the optimal model corresponding to the epoch with the lowest validation error was saved and used in the testing phase.

### 2.5. Performance Evaluation in the Testing Set

For each testing dataset, the reconstructed BP was generated with the optimal neural network resulting from the training and validation process. The mean square and mean absolute errors, as well as the coefficient of determination R^2^, were calculated across the entire reconstructed signal (see Figure 4). The differences in R^2^, mean squared error, and mean absolute errors between training and testing sets were statistically tested using a Mann–Whitney statistical test. The analysis of the dependency of the coefficient of determination in terms of demographic and health information for the test set was carried out using a linear mixed model (Appendix A). Statistical tests were not performed on the validation set because of its relatively small size.

Furthermore, systolic (SBP) and diastolic blood (DBP) pressure values were estimated by considering the maximum and minimum values of the BP waveform. Reference SBP and DBP were calculated on the reference BP waveform. Estimated SBP and DBP were calculated on the reconstructed BP waveform. For both reference and reconstructed BP waveforms, five-second window-based and recording-based SBP and DBP values were calculated.

Within a five-second window, the local maxima were detected by considering negative-going zero-crossings of the first derivative of the waveform. The detected negative-going zero-crossings were selected as local SBP values if they had an amplitude in the 100 mmHg to 150 mmHg range and were separated from adjacent detected local SBP values by at least 50 samples (i.e., 500 milliseconds). The positive-going zero-crossings were used to detect local minima; local DBP values were selected if they had an amplitude in the 60 mmHg to 100 mmHg and were separated from adjacent detected local DBP values by at least 50 samples. The amplitude ranges used to detect local maxima (100 to 150 mmHg) and minima (60 to 100 mmHg) were set considering the distributions presented in [23].

The recording-based SBP and DBP values were calculated as the mean value across all five-second windows in the recording. For both SBP and DBP, the agreement between the actual values (obtained from the reference BP waveform) and the estimated values (obtained from the reconstructed BP waveform) was evaluated using Bland–Altman plots [24].

## 3. Results

### 3.1. Training and Validation of the Neural Network

The training and validation learning curves are shown in Figure 5. The number of epochs (=47) for which the validation error was minimum is indicated with a vertical black dashed line. The training error exhibits an overall decreasing trend, but there are some regions where the error increases. The validation error indicates overfitting after 50 epochs.

### 3.2. BP Waveform Reconstruction Results

The root mean squared error, mean absolute error, and coefficient of determination R^2^ for the reconstructed BP waveform with respect to the reference BP are shown in Table 2. Examples of nine reconstructed BP waveforms (in orange), along with the corresponding BP reference (in blue), are shown in Figure 6. The analysis of the dependency of the coefficient of determination on gender, age, and health condition is presented in Appendix A.

### 3.3. Estimation of Systolic and Diastolic Blood Pressure

The Bland–Altman plots for the five-second window DBP and SBP are shown in Figure 7a,b, respectively. Each point represents a five-second window. The data from the windows of all recordings were pooled together to generate these plots. The upper and lower limits of agreement (LoA), which can be thought of as the upper and lower bounds for 95% of the differences with respect to the actual value, are indicated using horizontal, red, and dashed lines. The values in mmHg are also shown next to the lines on the right. For the sake of visualization, the units were omitted. In addition, the bias, which corresponds to the mean value of the difference between the estimated BP value and the reference one, is indicated by a horizontal, blue, and dashed line. The bias is shown as a blue text next to the corresponding horizontal line. For both DBP and SBP, the upper LoA exceeds 17 mmHg, and the lower LoA is at most −13.1 mmHg. The width of the limits-of-agreement interval is tighter for DBP (17.39 + 13.09 = 30.48 mmHg) compared to SBP (17.34 + 21.96 = 39.30 mmHg).

The Bland–Altman plots for the recording-based (i.e., the one that results from averaging the DBP and SPB values from all the five-second windows in a recording) DBP and SBP are shown in Figure 7c,d, respectively. Each point represents a recording which, as mentioned in the Methods section, is from different individuals compared to that included in the training and validation sets. Similar to the window SBP and DBP, the limits of agreement and bias are indicated in Figure 7c,d. The estimation of SBP and DBP using the mean values across all windows results in tighter limits of agreement. SBP is particularly interesting because it can be estimated with upper and lower limits of agreement of +3.5 mmHg and –8.0 mmHg, respectively.

## 4. Discussion

Unobtrusive monitoring of blood pressure during sleep may enable early evaluation of cardiac risk and, if required, opportune treatment administration. In this paper, the use of load-cell signals positioned under each leg of a bed for contactless estimation of blood pressure was investigated.

Instead of trying to directly estimate systolic and diastolic blood pressure, the BP waveform was first sequentially reconstructed using the data from a sliding matrix containing four rows corresponding to each load cell and with each row containing 0.5 s of load-cell data. By sliding the window throughout the entire duration of the load-cell signals, the entire BP waveform could be reconstructed except for the initial half a second. Systolic and diastolic blood pressure were then estimated as the maximum and minimum values from the reconstructed waveform.

Because of the high ability of neural networks to fit multi-variate regression problems, a deep neural network architecture composed of three recurrent LSTM layers and one dense layer was selected to reconstruct the next BP sample using the sliding matrix of load-cell data. The use of recurrent neural networks is common in computational physiology as they can effectively model temporal and nonstationary dynamics of physiological signals [26]. The partition of the data from the 38 individuals considered in this study into training, validation, and testing sets was made to ensure that the data from a given individual is considered in only one dataset. This is a strength of the procedure to test the model as it can provide testing results that more closely reflect the generalization across individuals.

The reconstructed BP waveforms in the test set have a high visual resemblance to the correspondent reference BP, as can be observed in Figure 6. The quantification of this visual impression (see Table 2) indicates that, for the test set, 61% of the variance in the BP waveform is captured by the model, which suggests that it can partially reconstruct the BP waveform.

The analysis of the dependency of the coefficient of determination for BP waveform reconstruction on demographic parameters (Appendix A) shows no statistically significant influence of gender, age, weight, height, or health condition. However, the small size of the testing dataset may have prevented the identification of factors affecting the accuracy of BP reconstruction.

It is important to note that the R^2^ and mean validation error for the training set are significantly higher and lower, respectively, than that for the testing set. This illustrates the challenge of generalizing neural network models across individuals. A further drop in performance can be expected when trying to generalize across hardware acquisition systems (e.g., different load-cell technology), bed types, or even pre-processing algorithms. A possible strategy to maintain high performance despite these challenges is to rely on transfer learning strategies [27], where the parameters of the neural network layers before the final output are adjusted to fit individual calibration data. In the case of blood pressure, this strategy can consist of recording a short segment of load-cell and ground-truth blood pressure data to personalize the model. The practical implementation of such an approach requires multiple considerations which are beyond the scope of this paper.

Reconstructed BP waveforms allow the estimation of systolic and diastolic blood pressure by detecting maxima and minima values on the BP waveform. Five-second rolling windows were used to limit the impact of outlier values and to evaluate the potential to estimate instantaneous (i.e., faster variations of SBP and DBP) and longer-term BP averages.

The width of the limits-of-agreement interval for the five-second-based estimates of DBP and SBP exceed 30 mmHg may seem at odds with the moderately high R^2^ and mean absolute error < 0.1 mmHg for the BP waveform reconstruction reported in Table 2. However, closer consideration of the results represented in Figure 6 suggests that errors in the reconstructed BP waveform tend to manifest in extreme (maxima or minima) regions, which are the ones that are relevant for SBP and DBP estimation. This observation can be considered to modify the loss function used to train the neural network such that extreme points have a higher weight in the reconstruction error. While this modification of the loss function may lead to higher SBP and DBP estimation accuracy, it may also degrade the reconstruction error of the BP waveform.

The Bland–Altman plot for window-based DBP estimation (see Figure 7a) suggests the presence of a proportional error (i.e., the difference between the estimated and actual DBP value increases as the actual value increases). The proportional bias may be addressed by transforming the dependent variable DBP using, for instance, the log function. The process of identifying a proper transformation of DBP is beyond the scope of this article, but it is among the optimizations planned for future work.

The estimation of SBP and DBP using the mean values across all windows in a recording provides much tighter limits of agreement (see Figure 7c,d). SBP is particularly interesting because it has lower (single-digit) limits of agreement. This level of accuracy, which matches the results reported in [10,11,12,13] detailed in the introduction section, may allow the detection of abnormally high or low BP values that are associated with cardiovascular risk. This is an encouraging result which also introduces an interesting research question for older-adult sleep. Indeed, periods of continuous sleep with a duration exceeding 8 min become less frequent as one ages [28]. While the focus here was on SBD and DBP estimation, reconstructing the entire BP waveform enables the calculation of additional hemodynamic parameters [29].

The research presented in this article has several limitations. The recordings from each user are short, 481 s on average, which makes it difficult to generalize the results to longer time periods which are relevant for sleep. Moreover, blood pressure has circadian variations [30] that are certainly not reflected in the data considered in this article. Additionally, BP may be influenced by contextual factors (medication, physical activity, or emotional state) before measurement that is not reported in the dataset.

The short duration of the recordings also prevents evaluation of the temporal coverage. Indeed, movements during bed presence can degrade the load-cell signal quality, making it difficult to reconstruct the BP waveform during those periods of low signal quality.

Another limitation is the absence of consideration of cardiovascular disorders in this research. Four participants reported a different heart condition but no qualification of severity. Arguably, individuals affected by cardiovascular conditions could benefit the most from the concept presented in this article and, therefore, future research must focus on specific patient groups, possibly starting with abnormally high or low blood pressure.

## 5. Patents

Patent application “Bed Having Features to Passively Monitor Blood Pressure”.

## Figures and Tables

**Figure 1 sensors-24-00096-f001:**
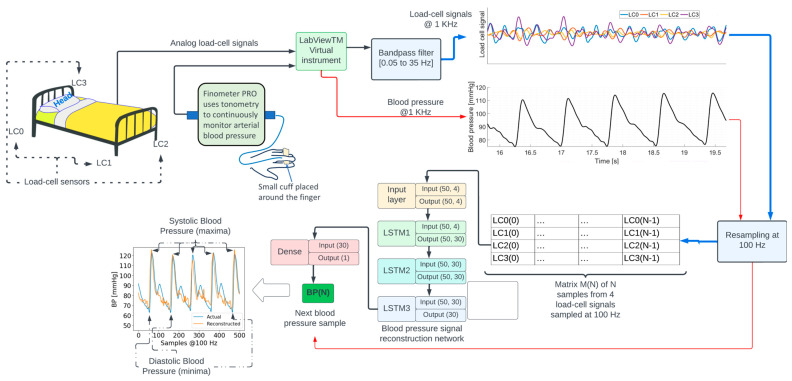
Overview of the signal processing and machine learning methods to reconstruct the blood pressure waveform.

**Figure 2 sensors-24-00096-f002:**
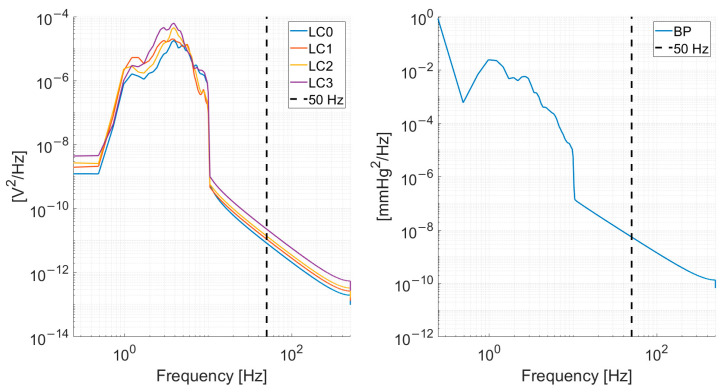
Mean (across all recordings from all subjects) power spectrum density for the load-cells (**left**) and blood pressure signal (**right**). A vertical dashed line at 50 Hz is shown to illustrate the rationale of subsampling at 100 Hz.

**Figure 3 sensors-24-00096-f003:**
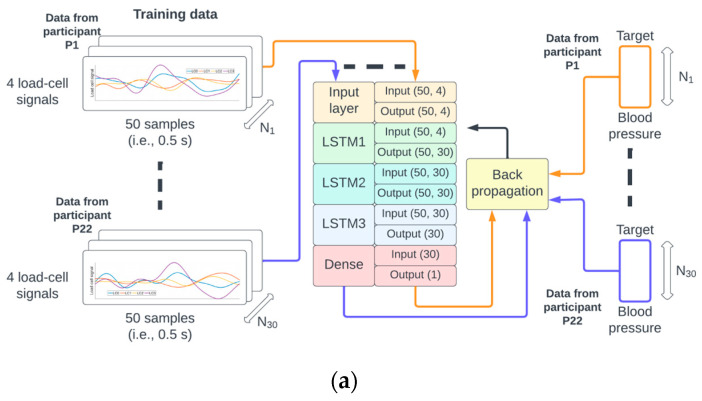
The identifiers from the 38 participants were randomly permuted and referred to as P1 to P38. The data from P1 to P22 were used for training (**a**), P23 to P28 for validation (**b**), and P29 to P38 for testing (**c**) of the neural network.

**Figure 4 sensors-24-00096-f004:**
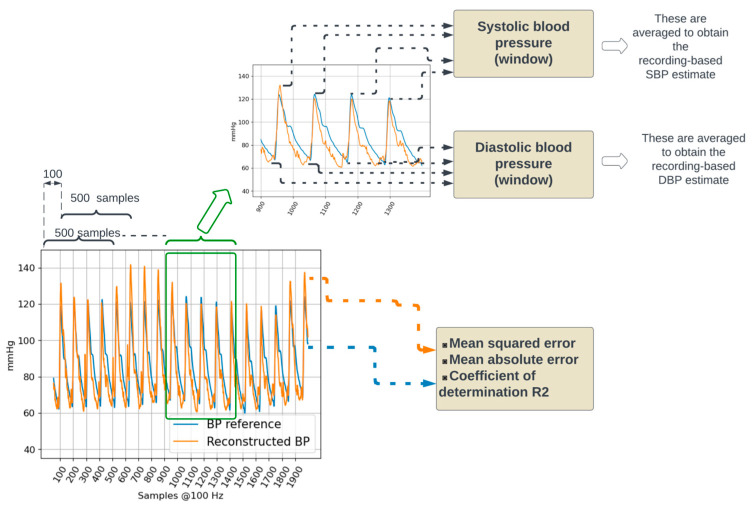
Performance evaluation. The reconstructed BP signal from a testing dataset is compared to the corresponding BP reference by calculating the mean squared and mean absolute errors. In addition, systolic and diastolic blood pressure values are calculated.

**Figure 5 sensors-24-00096-f005:**
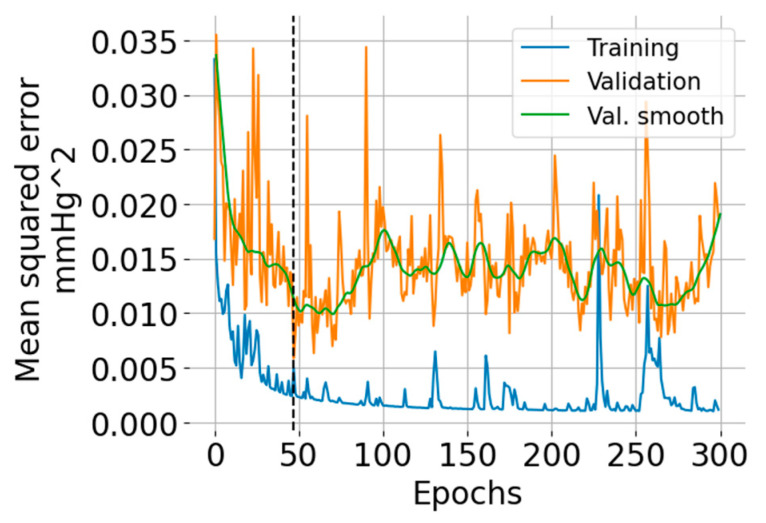
Training and validation learning curves showing the mean squared error versus the number of epochs. The green curve is a smoothed version (using the LOWESS algorithm [25]) of the validation error intended for ease of visualization.

**Figure 6 sensors-24-00096-f006:**
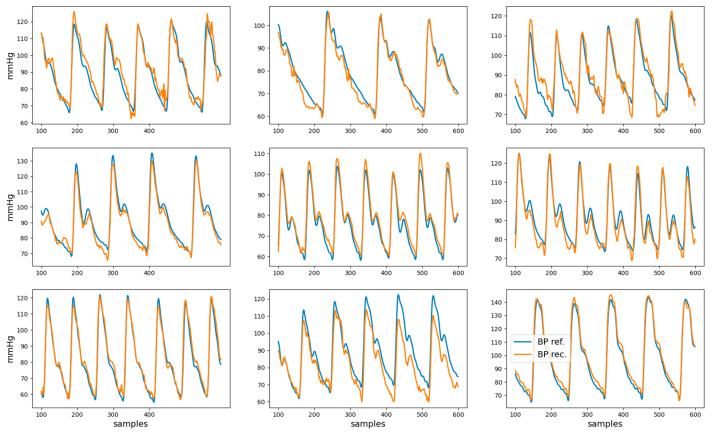
Examples of reference BP (blue) and reconstructed BP (orange) waveforms. These represent the data from second 1 to second 6 for nine different recordings.

**Figure 7 sensors-24-00096-f007:**
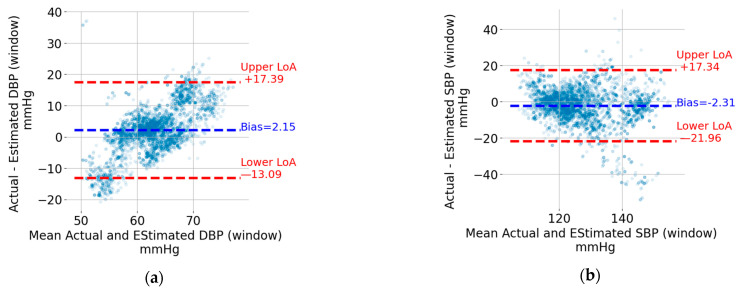
Bland–Altman plots for (**a**) DBP window, (**b**) SBP window, (**c**) DBP recording, and (**d**) SBP recording.

**Table 1 sensors-24-00096-t001:** Demographic information and recording duration for the participants included in the study.

	**Counts**
Gender	15 M, 23 F
Cardiovascular (CV) conditions	34 without CV conditions. 4 with CV conditions
	**Mean (Standard Dev.)**
Age (years)	34.6 (14.6)
BMI (Kg/m^2^)	26.1 (5.8)
Recording duration (seconds)	481.2 (26.5)
Count of inter-beat intervals	455.9 (95.1)

**Table 2 sensors-24-00096-t002:** BP waveform reconstruction results. The table entries are shown as mean (standard deviation).

	Training Set	Validation Set	Test Set	*p*-Value Train vs. Test
Mean squared error (mmHg^2^)	0.002 (0.001)	0.001 (0.0005)	0.012 (0.008)	<0.001
Mean absolute error (mmHg)	0.031 (0.007)	0.029 (0.006)	0.082 (0.033)	<0.001
Coefficient of determination R^2^	0.93 (0.040)	0.92 (0.013)	0.61 (0.221)	0.002

## Data Availability

The data presented in this study have been made openly available in IEEE DataPort at doi:10.21227/77hc-py84, reference [15].

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
