# Peer review of "Feasibility of Unobtrusively Estimating Blood Pressure Using Load Cells under the Legs of a Bed"

_sensors, 2023, doi:10.3390/s24010096_

Round 1

Reviewer 1 Report

Comments and Suggestions for Authors

The topic addressed in this manuscript would be of interest to readers of Sensors. The study examined the validity of using load cells under the legs of a bed in estimating blood pressure using neural network analysis. The study found acceptable results in using load cells to estimate blood pressure during sleep.

Major concern

The author performed a neural network analysis to showcase the validity of the load cells in estimating blood pressure. However, the current format of the discussion served as an extension of the results rather than a standalone discussion. In the discussion section, it presented information, such as equation and mean that could be better in the result section. Using the discussion section to further explain the usability of the load cells will help readers better understand the feasibility and usefulness of the results.

Author Response

The topic addressed in this manuscript would be of interest to readers of Sensors. The study examined the validity of using load cells under the legs of a bed in estimating blood pressure using neural network analysis. The study found acceptable results in using load cells to estimate blood pressure during sleep.

I would like to thank the reviewer for their comments. I believe that the journal Sensors is ideal to disseminate the ideas proposed in this manuscript. The ability to unobtrusively monitor blood pressure may be essential to promote better societal health outcomes. I am very appreciative of the reviewer comments which helped me reorganize the Results and Discussion sections in a more coherent manner.  

The author performed a neural network analysis to showcase the validity of the load cells in estimating blood pressure. However, the current format of the discussion served as an extension of the results rather than a standalone discussion. In the discussion section, it presented information, such as equation and mean that could be better in the result section. Using the discussion section to further explain the usability of the load cells will help readers better understand the feasibility and usefulness of the results.

I agree with the reviewer about the need to more coherently organize the Results and Discussion sections. The parts of the discussion that had the equation and other quantitative details were moved to the results section. The discussion section was also expanded to focus more on the possibilities enabled by this feasibility analysis. All of these changes have been highlighted in the revised manuscript.

Reviewer 2 Report

Comments and Suggestions for Authors

This manuscript falls under the Sensors scope and presents findings of research titled “Feasibility of unobtrusively estimating blood pressure using load-cells under the legs of a bed. The manuscript consists of 13 pages, 2 tables, and 7 figure. The paper presents interesting results as well as an inquisitive and reliable interpretation of the research results. The topic original and relevant in the field of study. The Abstract provides the highlights of the key contents of the main text. The Introduction provides enough background information to justify the study. The Results are consistent with the declared methodology, presented clearly enough, supported by the tables and figure. Researchers studied the ability to monitor blood pressure unobtrusively during sleep, which may promote prevention of cardiovascular diseases and timely administration of treatment, and concluded that the results match the state-of-the-art and may enable the detection of abnormally large or small variations in blood pressure which indicate cardiovascular health degradation. The methodology adequately described and conclusion consistent with the evidence and arguments presented. The references are appropriate and relevant to the research. However, minor typographical and grammatical errors need addressing.

Comments on the Quality of English Language

Minor typographical and grammatical errors need addressing.

Author Response

This manuscript falls under the Sensors scope and presents findings of research titled “Feasibility of unobtrusively estimating blood pressure using load-cells under the legs of a bed”. The manuscript consists of 13 pages, 2 tables, and 7 figure. The paper presents interesting results as well as an inquisitive and reliable interpretation of the research results. The topic original and relevant in the field of study. The Abstract provides the highlights of the key contents of the main text. The Introduction provides enough background information to justify the study. The Results are consistent with the declared methodology, presented clearly enough, supported by the tables and figure. Researchers studied the ability to monitor blood pressure unobtrusively during sleep, which may promote prevention of cardiovascular diseases and timely administration of treatment, and concluded that the results match the state-of-the-art and may enable the detection of abnormally large or small variations in blood pressure which indicate cardiovascular health degradation. The methodology adequately described and conclusion consistent with the evidence and arguments presented. The references are appropriate and relevant to the research. However, minor typographical and grammatical errors need addressing.

I would like to thank the reviewer for their positive comments about the ideas presented in this manuscript. I am hopeful that the ideas presented in this manuscript can contribute to the development of methods to unobtrusively and continuously monitor blood pressure. The audience of the journal Sensors is particularly well equipped to accomplish this prospect.  

I appreciate the reviewer comment about ensuring proper grammatical and typographic proof reading. The revised manuscript has the following highlighted changes: 1) The caption, text, and contents of Figure 3 were corrected for typos. 2) A confusing sentence on upper and lower LoA in Section 3.3. was simplified with the help of a scientist whose native language is English. 3) The equation and quantitative details in the Discussion section were moved to the Results section. 4) Several typographical and grammar errors have been corrected throughout the text (these are highlighted to facilitate the reviewer's perusal). 

Reviewer 3 Report

Comments and Suggestions for Authors

The paper aims to study the effectiveness of deep neutral network using the load-cell data in constructing the blood pressure waveform.

the paper is well organized and written and figure and tables are clearly presented. I have some comments

-       In Table 1: the unit of BMI is kg/m mot kg /m2?

-       In table 1: author presented the recording duration in second, was it during one night?

-       Regarding the number of patient chooses from training, testing and validation, is there any criteria used in the determination of the number?

-       The obtained value of the coefficient of determination R2 is 0.6! it means that the model partially construct the BP waveform!

-       Authors mentioned that ‘These results match the state-of-the- 21 art and may enable the detection of abnormally large or small variations in blood pressure which indicate cardiovascular health degradation.’ Reference for this info!

-       In Figure 3: Authors mentioned that they used only 38 patients, but they still mentioned the 40 participants.

-       In figure 3: why you mean by 50 samples (0.5 second)?

-       In figure 6: I can see that in some cases there is a difference between BP ref and BP recorded! Any explanation?? Related to gender, health situation…?

-       I have a questions regarding the situation of the participants before collecting the data, there were under the same circumstance, diet, medicine……

- Is there any similar model to compare the performance of your model with it?

Author Response

The paper aims to study the effectiveness of deep neutral network using the load-cell data in constructing the blood pressure waveform. The paper is well organized and written and figure and tables are clearly presented. I have some comments

I would like to thank the reviewer for their comments about this manuscript. In addition to providing precise details about parts of the manuscript that needed correction, the comments from the reviewer provided insightful suggestions to expand and strengthen the article. 

In Table 1: the unit of BMI is kg/m mot kg /m2?

The BMI unit is Kg/m2. The submitted version of the paper had the m2 with an exponential notation that may be confused with a footnote. In the revised version, "Kg/m2" was used in Table 1.

In table 1: author presented the recording duration in second, was it during one night?

The recordings were performed during a relatively short period of time. The average duration was 8 minutes. Clarifying this is important and prompted by the question from the reviewer, I emphasized the information about the duration in the second paragraph of the Materials and Methods Section (highlighted text). In addition, the duration  aspect is central to two limitation factors elaborated in the Discussion section (lines 302 to 310). 

Regarding the number of patient chosen from training, testing and validation, is there any criteria used in the determination of the number?

The partition percentages used for building the model and validating it were approximately set to 75% (28/38 participants) and 25% (10/38 participants) of the data respectively. The 75%, 25% partition was decided based on common machine learning recipes. Out of the 28 participants to build the model, the data from 22 of them were used for training and the rest for validation. This was clarified in Section 2.4 (changes have been highlighted).

The obtained value of the coefficient of determination R2 is 0.6! it means that the model partially construct the BP waveform!

I share the positive appreciation from the reviewer about the ability of the model to partially reconstruct the BP waveform. I slightly modified the text in lines 256 to 260 to emphasize the 0.61 value for the coefficient of determination R^2.

Authors mentioned that ‘These results match the state-of-the-art and may enable the detection of abnormally large or small variations in blood pressure which indicate cardiovascular health degradation.’ Reference for this info!

The list of references for this are presented of the manuscript:
[10] Seok et al, 2021 ; [11] chen et al. 2013, [12] Kim et al. 2018, and [13] Tian et al 2023. The limits-of-agreement for these are provided in lines 61 to 72 of the manuscript.
In order to acknowledge the need for additional validation and make it clear that this article focuses on feasibility,  I removed the "state-of-the-art" claim in the Abstract and mentioned that these results suggest the possibility of unobtrusive BP monitoring using force sensors under the legs of a bed.

In Figure 3: Authors mentioned that they used only 38 patients, but they still mentioned the 40 participants.

I would like to thank the reviewer for identifying this typo in Figure 3. The numbers were corrected in the caption and the contents of the Figure. The edits were highlighted to make it easier to identify them in the revised version.

 In figure 3: why you mean by 50 samples (0.5 second)?

50 samples is the equivalent of 0.5 seconds at a sampling frequency of 100 samples/s. This clarification was added to the figure to improve its readability.

In figure 6: I can see that in some cases there is a difference between BP ref and BP recorded! Any explanation?? Related to gender, health situation…?

This is a very interesting question that motivated the analysis of the dependency of the R2 for the BP waveform reconstruction on demographic factors including Gender, Age, Weight, Height, and Heart Condition. This analysis is reported in the Appendix A. The results show no statistically significant influence of any of those factors on the R2 value. However, the small size of the testing dataset may have prevented the detection of meaningful influence factors. The Methods, Result and Discussion sections were updated to report this analysis.

I have a questions regarding the situation of the participants before collecting the data, there were under the same circumstance, diet, medicine……

This is a relevant question that can be helpful to provide better result interpretability. This type of information was unfortunately not reported in the dataset. This was mentioned as a limitation in the discussion section

Is there any similar model to compare the performance of your model with it?

The precise concept about the BP waveform reconstruction has not been attempted before. However, for the estimation of SBP and DBP, several references exist. I mentioned those in the discussion section Lines 308 to 311. 
The specific limits-of-agreement for those references are detailed in the introduction section (lines 61 to 72). 
